# Developing Prediction Models Using Near-Infrared Spectroscopy to Quantify Cannabinoid Content in *Cannabis Sativa*

**DOI:** 10.3390/s23052607

**Published:** 2023-02-27

**Authors:** Jonathan Tran, Simone Vassiliadis, Aaron C. Elkins, Noel O. I. Cogan, Simone J. Rochfort

**Affiliations:** 1Agriculture Victoria Research, AgriBio Centre, AgriBio, Melbourne, VIC 3083, Australia; 2School of Applied Systems Biology, La Trobe University, Bundoora, VIC 3083, Australia

**Keywords:** cannabinoids, partial least square regression, partial least square discriminant analysis, principal component analysis

## Abstract

Cannabis is commercially cultivated for both therapeutic and recreational purposes in a growing number of jurisdictions. The main cannabinoids of interest are cannabidiol (CBD) and delta-9 tetrahydrocannabidiol (THC), which have applications in different therapeutic treatments. The rapid, nondestructive determination of cannabinoid levels has been achieved using near-infrared (NIR) spectroscopy coupled to high-quality compound reference data provided by liquid chromatography. However, most of the literature describes prediction models for the decarboxylated cannabinoids, e.g., THC and CBD, rather than naturally occurring analogues, tetrahydrocannabidiolic acid (THCA) and cannabidiolic acid (CBDA). The accurate prediction of these acidic cannabinoids has important implications for quality control for cultivators, manufacturers and regulatory bodies. Using high-quality liquid chromatography–mass spectroscopy (LCMS) data and NIR spectra data, we developed statistical models including principal component analysis (PCA) for data quality control, partial least squares regression (PLS-R) models to predict cannabinoid concentrations for 14 different cannabinoids and partial least squares discriminant analysis (PLS-DA) models to characterise cannabis samples into high-CBDA, high-THCA and even-ratio classes. This analysis employed two spectrometers, a scientific grade benchtop instrument (Bruker MPA II–Multi-Purpose FT-NIR Analyzer) and a handheld instrument (VIAVI MicroNIR Onsite-W). While the models from the benchtop instrument were generally more robust (99.4–100% accuracy prediction), the handheld device also performed well (83.1–100% accuracy prediction) with the added benefits of portability and speed. In addition, two cannabis inflorescence preparation methods were evaluated: finely ground and coarsely ground. The models generated from coarsely ground cannabis provided comparable predictions to that of the finely ground but represent significant timesaving in terms of sample preparation. This study demonstrates that a portable NIR handheld device paired with LCMS quantitative data can provide accurate cannabinoid predictions and potentially be of use for the rapid, high-throughput, nondestructive screening of cannabis material.

## 1. Introduction

*Cannabis sativa* has been used as a herbal medicine for thousands of years, with its first documented use on Egyptian Ebers papyrus dating back to the sixteenth century BC [1]. Today, medicinal cannabis has been shown to be beneficial in the treatment of neurological conditions, such as multiple sclerosis and epilepsy, and for the treatment of chronic pain [2]. While many pharmaceuticals are synthesised using laboratory reagents, cannabinoids are extracted from the inflorescences of cannabis plants, *Cannabis sativa* [2]. The two major cannabinoids of interest for their therapeutic effects, found in *C. sativa*, are cannabidiol (CBD) and delta-9-tetrahydrocannabinol (THC) [3,4]. CBD has been used in the treatment of intractable epilepsy, particularly in children [3], with research into its analgesic and anti-anxiety effects progressing [5]. THC is used in the treatment of multiple sclerosis (MS), spasticity, chronic pain and posttraumatic stress disorder (PTSD) [4,6,7]. Cannabidiolic acid (CBDA) and tetrahydrocannabinolic acid (THCA) are naturally synthesised molecules in the plant that are converted into the more active CBD and THC through a process of decarboxylation, typically achieved by heating cannabis inflorescences. CBD and THC do occur naturally in the plant but generally in lower levels as breakdown products from acid precursors. CBDA and THCA are the major cannabinoids in most plants, although their therapeutic potential is not as well studied as CBD and THC.

Some minor cannabinoids have been studied for therapeutic benefits. These include cannabinol (CBN) as an anti-inflammatory [8]; cannabigerolic acid (CBGA), cannabidivarinic acid (CBDVA) and cannabidivarin (CBDV) as anti-convulsants [9,10]; cannabigerol (CBG) has indications of being a treatment for MS and Parkinson’s disease (PD) [11]; tetrahydrocannabidivarin (THCV) has evidence of being an appetite suppressant and type-2 diabetes treatment [12,13]; and cannabichromene (CBC) has been studied as an anti-convulsant and in antitumor treatment [14,15]. Despite this, most manufacturers focus on products with high THC and/or CBD, meaning cultivators select plants high in THCA and/or CBDA.

Traditionally, the analysis of cannabinoids is completed using gas chromatography (GC) or liquid chromatography (LC) coupled with mass spectrometry (MS) as a detector; alternatively, flame-ionization detection (FID) is used for GC or an ultra-violet diode array detector (UV-DAD) is used for LC [16]. This process, although accurate and selective, is costly and slow compared with other techniques such as near-infrared (NIR) spectroscopy. NIR spectroscopy is a rapid, nondestructive but non-selective technique that can be used to rapidly scan cannabis in reflectance mode to produce NIR spectra [17,18,19,20,21]. These spectra can be used to make cannabinoid content predictions or distinguish between cultivars with significantly different chemotypic profiles. This can be achieved by developing prediction models from NIR spectra paired with high-quality quantitation data and multivariate statistical software. Sanchez et al. (2018) [22] developed cannabinoid content prediction models of CBDV (*R*^2^ = 0.92), Δ9-THCV (*R*^2^ = 0.87), CBD (*R*^2^ = 0.98), CBC (*R*^2^ = 0.93), Δ8-THC (*R*^2^ = 0.85), Δ9-THC (*R*^2^ = 0.90), CBG (*R*^2^ = 0.54) and CBN (*R*^2^ = 0.87) in finely ground, heat-treated cannabis inflorescences using data from two lab-based NIR instruments and with GC-FID data as a reference [22]. Deidda et al. (2021) [21] made prediction models for Δ9-THC using two handheld NIR instruments on three sample types, fresh whole inflorescences (*R*^2^ = 0.93, *R*^2^ = 0.73); coarsely ground (*R*^2^ = 0.76, *R*^2^ = 0.74); and sieved cannabis (*R*^2^ = 0.77, *R*^2^ = 0.93) samples, with UHPLC-UV data as a reference. Jaren et al. (2022) [20] used a handheld NIR to make prediction models for Δ9-THC (*R*^2^ = 0.77) and CBD (*R*^2^ = 0.77), with HPLC-DAD data as reference data, on heat-treated, finely ground material [20]. Yao et al. (2022) [19] used an FT-NIR handheld instrument to build predictive models for Δ9-THC (*R*^2^ = 0.91) and CBD (*R*^2^ = 0.93) concentrations in finely ground cannabis hemp samples. This research has demonstrated that NIR spectroscopy can be successfully used as a tool to predict cannabinoid concentration when accompanied by high-quality reference data. Apart from Sanchez’s study, these studies did not attempt to quantify the presence of cannabinoids such as CBDA, THCA, CBC, CBGA, CBG, CBDVA, CBDV THCV, THCVA, CBN, CBNA and CBCA. It is important for cultivators to obtain as much detailed information on the cannabinoid profile of their cannabis plants as possible, and information on concentrations is desirable. In addition, these studies utilise small sample sizes and lack chemovar variation. In statistics, it is always important to have a varied dataset with many samples to reduce any possibility of population bias. Lastly, the aforementioned work only focused on the main cannabinoids, CBD and Δ9-THC, in heat-treated material; without the decarboxylation step, CBDA and THCA are far higher in abundance compared with their derivatives [23]. In previous research, cannabis or hemp material was heat-treated before analysis, or quantitation was conducted using GC, which decarboxylates the acids and detects THC or CBD [18,20,22]. 

Birenboim et al. (2022) [17] developed prediction models for CBDA (*R*^2^ = 0.97), THCA (*R*^2^ = 0.95), CBD (*R*^2^ = 0.83), THC (*R*^2^ = 0.86), CBGA (*R*^2^ = 0.99), CBG (*R*^2^ = 0.72) and CBCA (*R*^2^ = 0.87) using an FT-NIR analyser to scan finely ground cannabis inflorescences and distinguished chemovars by classifying them as high THCA, high CBDA, high CBGA or ‘hybrid’ [17]. Some studies have also combined THC and THCA measurements to predict ‘total THC’ [21]. However, it may be useful to determine these individually since levels of decarboxylation can be an indication of maturation; as further research into the bioactivity of other cannabis constituents develops, quantifying amounts of the acids post-processing will become important. In addition, THC and THCA exposed to light and temperature are expected to degrade into CBN and CBNA over time through oxidation, and this is considered undesirable, as THC is a primary therapeutic target. These two cannabinoids should be of interest, as they may also serve as indicators of growth in long-term storage conditions. 

This current work describes the statistical analysis of inflorescences from 734 cannabis plants using NIR and LCMS data to develop different models using principal component analysis (PCA), partial least discriminant analysis (PLS-DA) and partial least squares regression (PLS-R). Prediction models for 14 cannabinoids (CBDA, THCA, CBD, THC, CBC, CBGA, CBG, CBDVA, CBDV THCV, THCVA, CBN, CBNA and CBCA) using 2 NIR systems and comparing finely and coarsely ground material is presented; finely ground material is more homogenous and reveals more surface area for the NIR to scan and, therefore, should provide better prediction models; conversely, coarsely ground material requires less effort but may produce weaker models, as coarsely ground is less homogenous. Ideally, if coarsely ground material can provide prediction models comparable to finely ground material, this would be a timesaving alternative. This work has applications in cultivation where the rapid assessment of cannabinoid profiles can be used to assess a harvest for regulatory compliance, product consistency or determining a market segment to provide the highest return on investment for the cultivator. Other potential uses include monitoring cannabinoid degradation over time and the impact of storage conditions and/or monitoring cannabinoid production during the vegetation and flowering stages. This technology and the associated prediction models could potentially be used by external quality assurance (QA) auditors as a rapid solution when inspecting cannabis plants to ensure cultivators adhere to their cultivation permits or other regulatory requirements during auditing.

## 2. Materials and Methods

### 2.1. Sample Preparation

All seeds were legally imported from Canada or generated by project activities, and all the work undertaken was performed under a Medicinal Cannabis Research Licence (RL011/18) and Permits (RL01118P6 and RLO1118P3) issued by the Department of Health (DoH), Office of Drug Control (ODC), Australia.

Plant growth and harvest conditions were as detailed in Naim-Feil et al. (2021) [24], where 734 cannabis plants consisting of 164 unique chemovars, with 4–5 biological repeats and were separated into 4 harvest groups (HG) according to harvest date: *n*= 479 for HG 1; *n*= 126 for HG 2; *n*= 78 for HG 3; and *n*= 51 for HG 4. All samples were freeze-dried for 48 h using a VirTis General Purpose Freeze Dryer (Scientific Products, Warminster, PA, USA) and then coarsely ground by placing the sample into a paper bag and hand crushing it until the material was approximately 3–5 mm in size prior to NIR scanning. Samples were then placed in liquid nitrogen for 1 min and ground to a fine powder using a SPEX SamplePrep 2010 Geno/Grinder (SPEX SamplePrep, Metuchen, NJ, USA) for 1 min at 1500 rpm. The fine powder was then subjected to the same NIR scanning methods as described below. A subsample of the ground powder was subjected to quantitation using LC-MS methods described by Elkins et al. (2019) [16]. A histogram plot of all cannabis samples detailing the cannabinoid range of each harvest group is provided in Appendix A.

### 2.2. Instrumentation and Parameters

Two near-infrared spectrometers were used to collect NIR scan data—a laboratory benchtop system and a handheld device. 

#### 2.2.1. FT-NIR Bruker MPA II

A laboratory benchtop NIR system was first used to acquire NIR spectral data: a Bruker MPA II–Multi Purpose FT-NIR Analyzer (Bruker Corporation, Billerica, MA, USA) equipped with TE-InGaAs detectors collected spectra in diffuse reflectance mode in a range of 11,500–4000 cm^−1^ (870–2500 nm). A 22 mm vial, 30-sample rotary scanner attachment was used to perform scans at a resolution of 16 cm^−1^ and 64 scans, with a measurement time of 25 s per individual sample. The ground cannabis samples were contained in 22 mm glass vials purchased from Bruker, ensuring that the sample saturated the bottom of the vial, and placed into the rotary scanner. Background was recorded automatically every hour with a gold-coated reference, and 4 QC samples were run every 30 samples to monitor any changes in signal in the instrument. The OPUS 8.2.21 software (Bruker Corporation, Billerica, MA, USA) was used for spectra data acquisition. To ensure reproducibility, scanning was performed in triplicate. The sample vials were agitated in between each scan to ensure homogeneity, and the data from the triplicate scans (total of 2202 individual scans) were averaged using Microsoft Excel 365 (Microsoft Corporation, Redmond, Washington, DC, USA) before data analysis. This wavenumber range contained complex spectral features exclusive to cannabinoids. In total, 734 individual cannabis samples were analysed.

#### 2.2.2. VIAVI MicroNIR Onsite-W

For comparative purposes, a VIAVI MicroNIR Onsite-W Spectrometer (VIAVI Solutions Inc., Scottsdale, AZ, USA) is a handheld portable NIR device that uses an InGaAs detector. Samples were decanted into sample cups/onto nitrogen-free weighing papers (Sigma-Aldrich, St. Louis, MI, USA) and scanned by directly pressing the detector against the cannabis sample. The detector was thoroughly cleaned using 80% methanol and a KimTech Kimwipe tissue (Kimberly-Clark, Irving, TA, USA) between each sample until there was no residue left. Scans were performed in triplicate, and the data were averaged as previously described. Data were collected between 10,526–6060 cm^−1^ (950–1650 nm). The device was configured in diffuse reflectance mode with an integration time of 12 milliseconds and 100 scan counts. Scanning was performed in standard laboratory conditions. Data acquisition was performed using MicroNIR Pro v3.0 (VIAVI Solutions Inc., Scottsdale, AZ, USA). In total, 730 cannabis samples were scanned.

### 2.3. Statistical Analysis

Bruker MPA II data were exported from the OPUS 8.2.21 software and imported into Microsoft Excel (Microsoft Corporation, Redmond, WA, USA), where the data from the triplicate scans were averaged. The averaged data were then imported into MATLAB 2022a (Mathworks, Natick, MA, USA) with PLS-Toolbox 9.0 (Eigenvector Research Inc, Manson, WA, USA) for analysis. The data were then trimmed to 9000–4000 cm^−1^ (1111–2500 nm). For statistical modelling, pre-processing was optimised using combinations of detrend, standard normal variate (SNV) and normalization, and 1st-order and 2nd-order derivatives were selected based on the best *R*^2^ prediction whilst maintaining low prediction bias. PCA was performed to check data quality and explore trends within the sample population. PLS-DA models were used to further predict specific class information such as CBDA and THCA ratios, and PLS-R models were used to create predictive models for cannabinoids: CBDA, THCA, CBD, THC, CBC, CBGA, CBG, CBDVA, CBDV THCV, THCVA, CBN, CBNA and CBCA. Both PLS techniques used LCMS quantitative data as independent variables and NIR data from both the Bruker MPA II and VIAVI MicroNIR Onsite-W as dependent variables. Separate regression models were developed for each compound, and the method with the highest *R*^2^ prediction value was selected. Venetian blinds cross-validation was used with 10 splits and blind thickness set to 1. 

When generating calibration and validation sets for PLS-R and PLS-DA models, a 75:25 split of the data was performed on the sample set. The Kennard–Stone algorithm was used to ensure the data selected for the calibration set were uniform and representative of the whole dataset, while the validation set contained samples that were interior and exterior to the calibration set. The algorithm achieves this by randomly seeking two samples within the dataset with the largest distance measure using either Euclidean or Mahalanobis distance [25]. This eliminates extrapolation in the calibration model when applied to the validation test. To test for statistical significance, permutation tests (*n* = 50) were carried out on all PLS models (Wilcoxon, Sign Test and Rand *t*-test).

## 3. Results and Discussion

### 3.1. Principal Component Analysis of NIR Data

A PCA was performed on the dataset to identify trends and assess data quality. Initial analysis of the data showed that the triplicate scans clustered well, highlighting good reproducibility (refer to Appendix A), and no spectra had to be removed from the dataset (Figure 1a,b). The average spectra for each sample were thereby used for all future models.

The analysis of the Bruker MPA II NIR spectra from the 734 samples showed that there were 3 major clusters evident across principal component (PC) 1, accounting for 65.39% of the variation within the dataset (Figure 1c). It was postulated that this clustering was related to cannabinoid content, particularly between high-CBDA, high-THCA and even-ratio chemovars (refer to Appendix A); this was proven to be correct once the NIR dataset was annotated according to the high CBDA, high THCA and even ratio. The PCA was repeated using the MicroNIR NIR data and the same high-CBDA and -THCA and even-ratio class information (Figure 1d). Although clustering in the chemovars was still apparent (38.71% of the variation within the dataset was distributed across PC1), even-ratio and high-CBDA chemovars had some overlap. PCA are typically performed to assist in building calibration and validation datasets by identifying outliers that may influence the performance of the model while exploring variations between samples [20,22]. PCA models performed by Sanchez et al. [22] on their cannabis dataset showed clustering; however, there was no elaboration on what the principal components or clustering represent, so it is important to investigate all trends and clusters that occur. Conversely, PCA models performed by other researchers have been used to identify distinct clustering within their datasets, and they have classified their samples as high CBDA, high THCA, high CBGA or ‘hybrid’. In addition, Birenboim et al. [17] (2022) were able to assign clusters to individual cultivars with high accuracy. Given the PCA showed clustering based on the chemotype, supervised PLS-DA models were investigated to determine how accurately the high-CBDA, high-THCA, and even-ratio chemovars could be predicted from the spectra.

### 3.2. Partial Least Squares Discriminant Analysis (PLS-DA) Modelling

PLS-DA models were able to divide the cannabis dataset into the three previously described categories: the even-ratio, high-CBDA and high-THCA classes (Figure 2a–c). High CBDA was defined as classes that had a concentration ratio of CBDA to THCA higher than 6:1; high THCA had a concentration ratio of less than 1:6; and even ratio had concentration ratios between 6:1 and 1:6 (CBDA:THCA) (refer to Appendix A).

#### 3.2.1. FT-NIR Bruker MPA II Data

The dataset scanned with the Bruker MPA II had a total of 734 finely ground samples. Sensitivity prediction and specificity prediction values were one and equal for the high-CBDA and high-THCA predictors (Table 1). Typically when sensitivity and specificity values are equal to each other, this is an indicator of a highly accurate PLS-DA model [26]. Our prediction models showed a classification accuracy of 100% (Class Error Pred. = 0%) for both high-CBDA and high-THCA models and 99.4% (Class Error Pred. = 0.7%) for even-ratio models. These models showed excellent results in predicting high-CBDA, THCA and even-ratio classes. 

Birenboim et al. [17] (2022) created PLS-DA models to make classification models of high-THCA, high-CBDA, high-CBGA and ‘hybrid’ classes, and the results provided accurate classifications based on a cannabis dataset of 325 cannabis samples relating to 15 unique chemovars. As noted by Birenboim, there were only 10–30 biological repeats per chemovar, meaning there was little genetic variation [17]. However, the present study has a cannabis dataset consisting of 734 cannabis samples with 164 unique chemovars with 4–5 biological repeats, resulting in larger genetic variation and the elimination of population bias due to the high sample number and different chemovars available whilst retaining sample repeatability. This sample size is certainly the largest and most complex, leading to the most robust prediction model currently developed.

#### 3.2.2. VIAVI MicroNIR OnSite-W Handheld Data

The PLS-DA model using the MicroNIR data had a total of 730 samples for each cannabinoid. PLS-DA modelling for MicroNIR used the same definitions for even ratio, high CBDA and high THCA (Figure 3a–c). High THCA had a classification accuracy of 100% (Class Error Pred., 0%); high CBDA was at 94.5% (Class Error Pred., 5.5%); and even ratio was at 83.1% (Class Error Pred., 16.9%) (Table 2). Whilst the predictions for high-THCA chemovars were accurate, reduced accuracy for the high-CBDA and even-ratio predictions were possibly due to the limitation of the resolution and range of the Micro NIR. Duchateau et al. used handheld NIR devices to create prediction models and focused on discriminating between ‘illegal’ (THC content > 0.2% *w*/*w*) and ‘legal’ cannabis. From a law enforcement perspective, this is useful, but this research is not able to truly define chemovars as high-THCA or high-CBDA classes [27]. The purpose of comparing the two instruments in the current study was to identify a rapid, field-deployable and inexpensive cultivar assessment tool for use in the cannabis industry in both commercial and research settings. Limited research has used MicroNIR and other handheld devices to develop prediction models, and researchers have typically built PLS-R models [20,21,22] over PLS-DA models. However, when considering the ability to rapidly quantify cannabinoid content, it largely rests on user requirements, for example, an exact concentration, which provides a numerical value or a classification on whether it is a high-THCA or high-CBDA chemovar [21]. Classification prediction models may be more useful than rapid cannabinoid assessments for cultivators where precise cannabinoid quantification values may not be the priority, as opposed to guaranteeing whether a cannabis plant is a high-THCA or high-CBDA chemovar.

### 3.3. Partial Least Squares Regression (PLS-R) Modelling

Creating prediction models using partial least squares regression will provide prediction values ranging from *R*^2^ 0 to 1. This value determines the accuracy of a prediction model. According to Williams et al. (2019) [28], prediction models can be classified as having a poor correlation (*R*^2^ = 0.26–0.49); predictions adequate for rough screening (*R*^2^ = 0.50–0.64); predictions adequate for screening and approximate predictions (*R*^2^ = 0.65–0.81); good predictors for applications such as research but not quality assurance (QA) (*R*^2^ = 0.82–0.90); great predictions that can be used in QA settings (*R*^2^ = 0.91–0.97); and excellent predictors (*R*^2^ > 0.98), which can be used in any application [28]. The ratio of performance of deviation (RPD) is a value typically reported as a measure of the goodness of fit of a prediction model; this has not been reported in this paper’s findings, as Minasny et al. (2013) [29] mentions that RPD and *R*^2^ values are the same measures. The current literature echoes this sentiment, as Martin (2022) [30] deems the RPD measure an inadequate indicator of a precise prediction model, instead preferring to use high *R*^2^ values and a ratio between the standard error prediction (SEP) and the standard laboratory error (SEL) as a measure of a strong prediction model, where the closer SEP is to SEL, the greater the precision of the model. SEL values in this study were reported (refer to S5) when a majority of cannabinoid SEL values were close to SEP values, providing good–excellent precision.

#### 3.3.1. FT-NIR Bruker MPA II Data

This dataset consisted of 734 samples that were prepared according to Section 2.1, Sample Preparation. The scanned samples were finely ground cannabis powders.

Different pre-processing parameters were applied to optimise the prediction of 14 cannabinoid concentrations (Table 3, Figure 4). The predictions for CBDA and THCA were highly accurate, with a regression value *R*^2^ of 0.98 for both cannabinoids. This was followed by THC, with an *R*^2^ value of 0.93, and CBD at 0.89. CBN and CBNA had *R*^2^ values of 0.80. The *R*^2^ values for the CBDVA, CBDV, CBGA and CBCA prediction models ranged from 0.53 to 0.61; according to Williams et al. (2019) [28], these values are still adequate for the rough screening of cannabinoids from dried finely ground cannabis inflorescences. The models for CBC, CBG, THCV and THCVA provided *R*^2^ values between 0.34 and 0.46, indicating a poor correlation between the predicted and measured groups. 

Throughout this study, all PLS-R models benefited from a variety of different data pre-treatment parameters to explore the best prediction results. Applications included SNV, second derivative, detrend and normalisation. The Bruker MPA II FT-NIR spectrometer produced high-quality NIR spectral data with good resolution. The CBDA, THCA and THC models had high values (*R*^2^ > 0.91) and are ideal for QA applications. The CBD model is suitable for research applications (*R*^2^ = 0.82–0.90). The model for CBNA and CBN is suitable for approximate predictions and screening (*R*^2^ = 0.65–0.81), whilst the CBCA, CBDVA, CBDV and CBGA models are only strong enough for rough screening (*R*^2^ = 0.50–0.64). The CBC, CBG, THCV and THCVA models had poor correlation (*R*^2^ < 0.49), most likely due to a poor abundance of the given cannabinoid. 

Recent research has developed cannabinoid prediction models using heat-treated cannabis inflorescences [20,21,22]. Because of this, the researchers were not able to quantify CBDA and THCA, which are the most abundant cannabinoids produced, and, therefore, did not analyse the original metabolic profile of the cannabis inflorescences prior to decarboxylation. Alternatively, Birenboim et al. analysed cannabis inflorescence samples without heat treatment and developed robust prediction models for CBDA (*R*^2^ = 0.97), THCA (*R*^2^ = 0.95), THC (*R*^2^ = 0.86) and CBD (*R*^2^ = 0.83) [17]. This is similar to the present study; however, improved *R*^2^ values were developed using the prediction models for CBDA (*R*^2^ = 0.98), THCA (*R*^2^ = 0.98), THC (*R*^2^ = 0.93) and CBD (*R*^2^ = 0.89). This is likely because our dataset is much larger, with 734 samples and 164 unique chemovars with 4–5 biological repeats, which provided a more robust model. In addition, prediction models were developed for CBN (*R*^2^ = 0.80), CBNA (*R*^2^ = 0.80), CBDVA (*R*^2^ = 0.60) and CBDV (*R*^2^ = 0.59) suitable for screening purposes [28]; this is novel for the cannabis industry, as these cannabinoids do not have developed PLS-R models for concentration prediction.

It is likely that higher *R*^2^ values can be achieved when there is a high abundance of the target cannabinoid present; typically, CBDA and THCA are the most abundant, followed by CBD and THC. Hence, their high *R*^2^ values are relative to the other cannabinoids present. A high cannabinoid concentration in a sample will enable the NIR to detect this cannabinoid with greater intensity; this intensity then is paired with quality LCMS data and processed using statistical software to apply machine learning algorithms identifying correlations and patterns. This is not ideal for low-concentration cannabinoids such as CBC, which are overshadowed by higher-concentration cannabinoids such as CBDA; because of this, their models may result in poor prediction.

It is also important to include samples that have varying concentrations of cannabinoids to ensure an unbiased population. Having a model that only has extremely high or low values of a cannabinoid may yield false high *R*^2^ prediction values, as the data will contain a population bias that will overfit the data.

#### 3.3.2. VIAVI MicroNIR OnSite-W Handheld Data

This dataset consisted of 730 samples prepared according to Section 2.1, Sample Preparation. The scanned samples were finely ground cannabis powders.

The VIAVI MicroNIR was utilised in this study to explore the benefits of using a handheld device as opposed to a benchtop instrument. Although it has a shorter wavelength range of 950–1650 nm (10,526–6060 cm^−1^) and fewer NIR datapoints compared with the Bruker MPA II, comparable cannabinoid predictive results were achieved (Table 4, Figure 5). Prediction models for both CBDA and THCA returned an *R*^2^ prediction value of 0.98, excellent for any application including research and QA, followed by CBD (*R*^2^ = 0.80) and THC (*R*^2^ = 0.75), CBNA (*R*^2^ = 0.76); CBN (*R*^2^ = 0.71); and CBCA (*R*^2^ = 0.66), suitable for screening and some approximate calculations. Prediction models for CBDVA (*R*^2^ = 0.55) and CBDV (*R*^2^ = 0.51) are adequate for rough prediction, whilst the models developed for CBC, CBGA, CBG, THCV and THCVA perform poorly, ranging from *R*^2^ 0.21 to 0.38.

Deidda et al. [21] (2021) used the MicroNIR to create prediction models for THC (*R*^2^ = 0.77) on 26 finely ground cannabis inflorescence samples. The study illustrated good preliminary findings but lacked a large dataset, and the source of the cannabis samples was vague. In addition, the focus on only THC is not sufficient, even from a law enforcement perspective, as not all THCA is decarboxylated into the active THC. Therefore, crucial information regarding cannabinoids that can be easily heated into its psychoactive derivative is lacking. Jaren et al. used a Luminar 5030 miniature handheld NIR to make prediction models for CBD (*R*^2^ = 0.77) and Δ9-THC (*R*^2^ = 0.77), coupled with HPLC-DAD data as reference data, on 35 heat-treated and finely ground cannabis hemp samples [20]. Yao et al. used an FT-NIR handheld instrument to build predictive models for Δ9-THC (*R*^2^ = 0.91) and CBD (*R*^2^ = 0.93) concentrations in 91 finely ground cannabis hemp samples [19]. Yao et al. and Jaren et al. only focused on CBD and THC and not their acidic forms; it is important to gather information about CBDA and THCA when determining cannabinoid profiles. In addition, these authors only looked at cannabis hemp samples that contained 0.3% or less THC. In the present study, both CBD and THC, as well as their acidic forms, CBDA and THCA, prediction models were developed along with six additional cannabinoids (CBN, CBDVA, CBDV, CBGA, CBNA and CBCA), some of which can be used for general screening or QA purposes. In addition, a sample size of 730 was used that contained 164 unique chemovars, reducing population bias. The genetic variation in the sample size is a great advantage over the two papers that only focused on cannabis hemp samples.

It is interesting that CBDA and THCA prediction models performed well using the MicroNIR compared with the Bruker MPA II considering its limited range and resolution. The MicroNIR is more than adequate and still able to perform excellent predictions for CBDA and THCA, rivalling the Bruker MPA II whilst being a smaller, rapid and portable solution. Considering that the cannabis industry still largely focuses on CBDA and THCA when producing cannabis products, this device can be successfully deployed in regulatory affairs and QA audits, where rapid inspection can classify a cannabis plant as a high-CBD or high-THC chemovar based on the CBDA and THCA predictions. 

The CBD, THC, CBN, CBNA and CBCA prediction models using the MicroNIR did not perform as well when compared with the Bruker MPA II but are still within ranges for screening and provide adequate insight into the chemotypic profile of the plant. Since CBNA and CBN are both oxidation products from THCA and THC and have *R*^2^ prediction values >0.70, this technique can provide insight into the quality of inflorescence material and the impact of storage conditions, as it slowly oxidises due to air, temperature and light, and it can be used as a rapid application for monitoring cannabinoid content for long-term storage [31,32].

### 3.4. Cannabinoid Correlation Matrix

Correlation coefficients were obtained for 14 cannabinoids (Table 5) using available LCMS data (refer to S6) to see the statistical relationship between each cannabinoid compound, where values nearing 1 indicate a positive correlation, values nearing −1 indicate a negative correlation and values nearing 0 indicate no correlation. High negative correlation coefficients were observed for CBDA and THCA (−0.80), where high levels of one cannabinoid indicate lower levels of the other. The same was seen for CBD and THC; however, the negative correlation is weaker at −0.50. 

Minor cannabinoids may have strong prediction models due to strong positive correlations to more abundant cannabinoids that have strong *R*^2^ prediction models. In the case of CBDA and CBD, the CBDA *R*^2^ prediction is 0.98, and the CBD *R*^2^ prediction is 0.89, despite CBD being lower in concentration compared with CBDA, with a correlation coefficient of 0.79. This observation is repeated for THCA (*R*^2^ = 0.98) and THC (*R*^2^ = 0.93), with a correlation coefficient of 0.66, where THC is lower in concentration than THCA. CBDA-CBD (0.79), THCA-THC (0.66) and CBDVA-CBDV (0.82) have positive correlations, which can be explained by the decarboxylation pathway of their acid forms to their neutral forms. THC (*R*^2^ = 0.93) and CBN (*R*^2^ = 0.80) also have a high correlation coefficient of 0.78, and this may be related to the oxidation pathway, where, over time, THC degrades into CBN; this degradation pathway and correlation also apply to THCA and CBNA. CBDA also correlates positively with CBDVA, CBDV and CBCA. Our findings are consistent with correlation matrices of cannabinoids found in the previous literature, where CBDA, CBD and CBCA are positively correlated with each other, whilst THCA and THC are positively correlated [33]. As with the present study, the previous literature also reports a negative correlation between CBDA and THCA. Birenboim et al. created a correlation matrix, showing a positive correlation between certain abundant cannabinoids and terpenes [33]. Overall, it seems that minor cannabinoids that have a positive correlation with more abundant cannabinoids (due to sharing a chemical pathway) have a positive influence on the *R*^2^ prediction.

### 3.5. Finely Ground vs. Coarsely Ground Material Using Bruker MPA II FT-NIR

A comparison was made of finely ground (Table 6) and coarsely ground (Table 7) inflorescences using the Bruker MPA II and scanning 479 samples (HG1). The aim of this was to see if the results were comparable between the two sample forms, as coarsely ground inflorescences will require less effort and preparation time. Prediction values for THCA (0.98 vs. 0.94), CBD (0.88 vs. 0.80) and THC (0.91 vs. 0.83) were better in all instances for finely ground inflorescences when compared with coarsely ground inflorescences. This was expected, as finely ground inflorescences have less area between the particles, increasing the scannable surface error of the sample; the sample is also more homogenous than that of the coarsely ground one. However, CBDVA (0.44 vs. 0.53), CBGA (0.43 vs. 0.51) and CBG (0.28 vs. 0.51) prediction values performed better in coarsely ground inflorescences; CBC (0.28 vs. 0.34), THCV (0.14 vs. 0.43) and THCVA (0.09 vs. 0.37) all performed better in the coarsely ground material form but did not have *R*^2^ values above 0.50, which cannot produce a model adequate for rough screening. The reason for this anomaly is unclear, and further investigation is required. CBDA (0.99 vs. 0.98), CBN (0.71 vs. 0.69), CBCA (0.60 vs. 0.59) and CBNA (0.79 vs. 0.79) saw negligible differences in *R*^2^ prediction values. CBDV (0.93 vs. 0.95) was also negligible; however, the *R*^2^ prediction values were unusually high compared with CBDV in Table 3; it is to be noted that this cannabinoid model has a population bias of high and low concentrations only influencing the model accuracy. More datapoints are required to correct this and improve the accuracy of the model. 

Deidda et al. [21] (2021) developed prediction models for THC and compared whole inflorescences, coarsely ground inflorescences and finely ground (sieved) inflorescences, with inconsistent results and found that when using the NIR-S-G1, finely ground inflorescences provided the most accurate prediction (*R*^2^ = 0.73, 0.74, 0.93); however, when compared with the MicroNIR, the whole inflorescence was the most accurate (*R*^2^ = 0.93, 0.76, 0.77). These results are contradictory to the current studies’ findings when comparing coarsely ground and finely ground inflorescences using the MicroNIR and Bruker MPA II, where the finely ground inflorescences provided the most accurate prediction.

Su et al. [18] (2022) compared non-decarboxylated whole hemp and ground hemp and developed prediction models using a Perten DA7250 NIR Analyzer for five cannabinoids: CBD (*R*^2^ = 0.89, 0.85); THC (*R*^2^ = 0.11, 0.25); CBN (*R*^2^ = 0.03, 0.38); CBG (*R*^2^ = 0.43, 0.03); and CBC (*R*^2^ = 0.79, whole inflorescence only). Although this study showed strong predictors for CBD and CBC, THC and CBN were weak, with almost no correlation, which is expected, as they were only looking at high-CBD cannabis hemp. When comparing physical sample states, whole hemp samples produced better *R*^2^ values than those of ground hemp samples, which is contradictory to Deidda’s NIR-S-G1 results. 

Overall, the current work found that scanning coarsely ground material provided quite comparable results to finely ground material, with the added benefit of saving time on sample processing. This is ideal for screening purposes, as it is cost-effective and would require less manual labour; in an industry setting where cultivators grow and harvest cannabis plants are at a large scale, this technology would be greatly valued. Depending on the use case, it may be better to opt for finely ground material to achieve the highest *R*^2^ prediction values for QA scenarios (*R*^2^ > 0.91), such as CBDA, THCA and THC (Table 6); however, coarsely ground cannabis material can also achieve results suitable for QA purposes for CBDA and THCA predictions (Table 7). From an industry perspective, this would be most desirable, as minimum standards for QA are being met whilst reducing workload.

## 4. Conclusions

Predictive models for the quantitation of 14 cannabinoids (CBDA, THCA, CBD, THC, CBC, CBGA, CBG, CBDVA, CBDV THCV, THCVA, CBN, CBNA and CBCA) and the characterisation of high-CBDA, high-THCA and even-ratio chemovars in cannabis inflorescence samples were developed using a combination of NIR techniques and LCMS data to build PLS-R and PLS-DA models. Two different NIR instruments were used, and it was found that the Bruker MPA II produced better results overall. However, the MicroNIR provided high-quality predictions for the main cannabinoid precursors (CBDA and THCA) and is accurate for several other cannabinoids (CBD, THC, CBNA, CBN and CBCA), despite being outperformed by the benchtop Bruker MPA II. Although the Bruker MPA II has a 30-vial rotary attachment that improves high throughput (scanning 30 samples in 15 min), the MicroNIR has the benefits of portability, rapid acquisition and a lower cost. PLS models with high predictive ability for CBDA and THCA were developed for both techniques and indicated that the abundance of a compound is one of the main influences in producing prediction models with high *R*^2^ values, and minor cannabinoids that correlate strongly with these major molecules also have a higher predictive *R*^2^ even at low concentrations. Further research is required to investigate potential correlations between major and minor cannabinoids and terpenes, potentially providing prediction models for terpene content. The analysis of coarsely or finely ground material was consistent, with coarsely ground material producing slightly less accurate prediction models compared with finely ground material, making coarsely ground material a suitable alternative if rapid, high-throughput screening is preferred over high-quality predictive models made for quality assurance or research. This process can be improved further if non-ground whole inflorescences can produce similar results, as the timesaving would be greater.

## Figures and Tables

**Figure 1 sensors-23-02607-f001:**
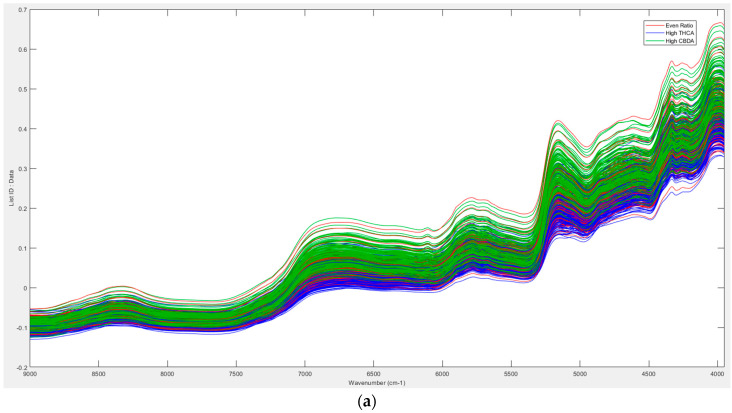
(**a**) Near-infrared spectra of all samples using the Bruker MPA II FT-NIR spectrometer in the 9000 cm^−1^ to 4000 cm^−1^ range. Green line: high CBDA; blue line: high THCA; red line: even ratio. (**b**) Near-infrared spectra of all samples using the handheld VIAVI MicroNIR Onsite-W in the 950 nm to 1650 nm range. Green line: high CBDA; blue line: high THCA; red line: even ratio. (**c**) Scores plot of two principal components of the entire cannabis sample set (averaged data, *n* = 734) using the Bruker MPA II FT-NIR spectrometer (9000 cm^−1^ to 4000 cm^−1^ range). The three major clusters across PC1 were assigned as high THCA (blue triangles), high CBDA (green squares) and even ratios (red diamonds). (**d**) Scores plot of two principal components of the entire cannabis sample set (averaged data, *n* = 730) using the VIAVI MicroNIR Onsite-W. The three major clusters across PC1 were assigned as high THCA (blue triangles), high CBDA (green squares) and even ratios (red diamonds).

**Figure 2 sensors-23-02607-f002:**
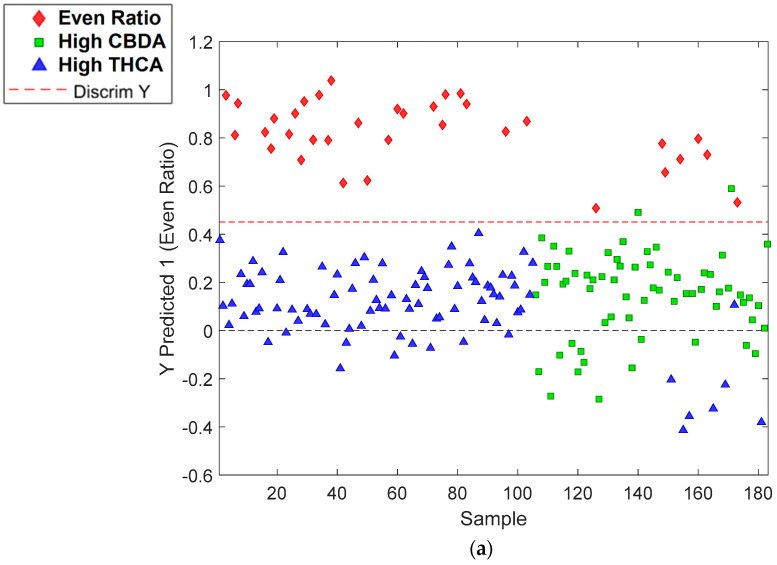
The classification of (**a**) even-ratio; (**b**) high-CBDA; and (**c**) high-THCA classes using data obtained from the Bruker MPA II instrument.

**Figure 3 sensors-23-02607-f003:**
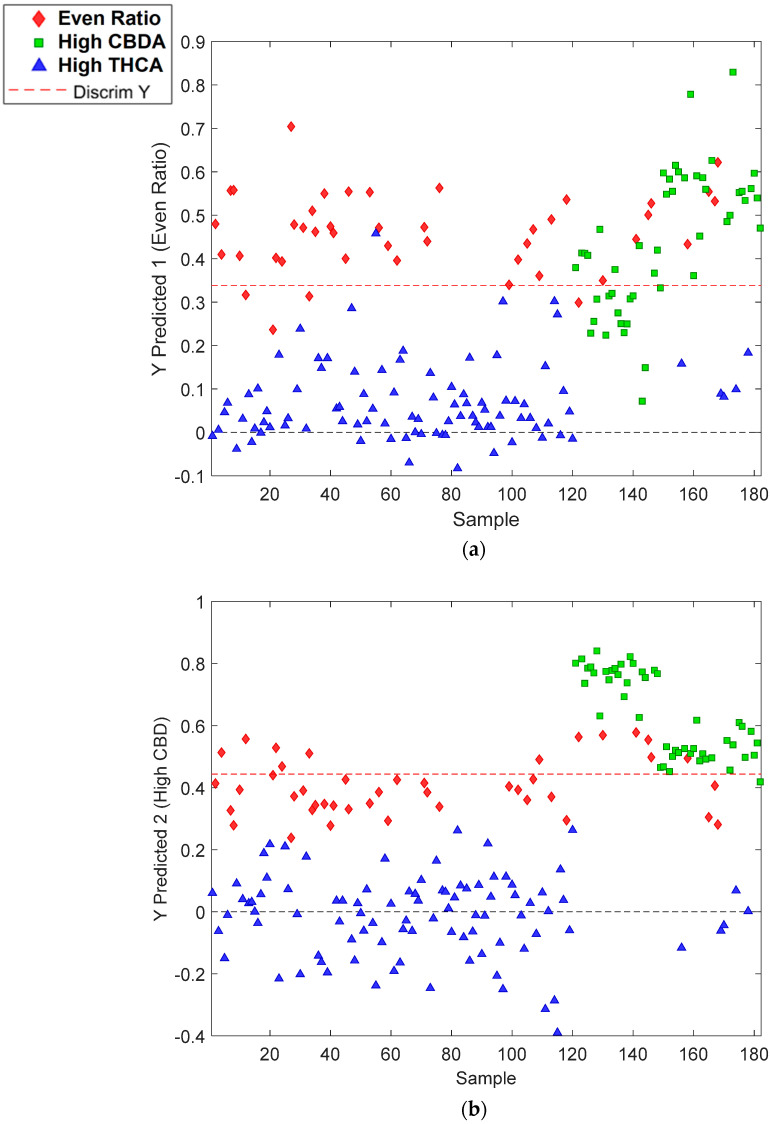
The classification of (**a**) even-ratio; (**b**) high-CBDA; and (**c**) high-THCA classes using the VIAVI MicroNIR OnsiteW instrument.

**Figure 4 sensors-23-02607-f004:**
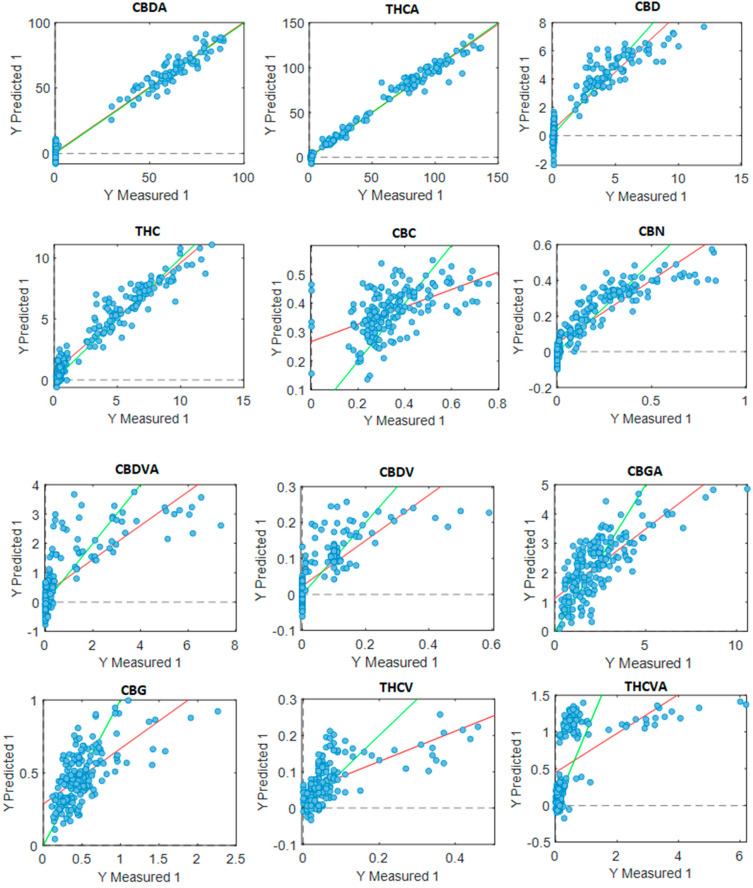
Plot of measured values versus the predicted values of cannabinoids in the validation dataset with data obtained from the Bruker MPA II instrument processed from the PLS_toolbox using the PLS-R tool. Cannabidiolic acid (CBDA), tetrahydrocannabinolic acid (THCA-A), cannabidiol (CBD), tetrahydrocannabinol (THC), cannabichromene (CBC), cannabinol (CBN), cannabidivaric acid (CBDVA), cannabidivarin (CBDV), cannabigerolic acid (CBGA), cannabigerol (CBG), tetrahydrocannabidivarin (THCV), tetrahydrocannabidivarinic acid (THCVA), cannabinolic acid (CBNA) and cannabichromenic acid (CBCA). Units are expressed in mg/g. Green line: line of best fit from reference data; red line: line of best fit from predicted data.

**Figure 5 sensors-23-02607-f005:**
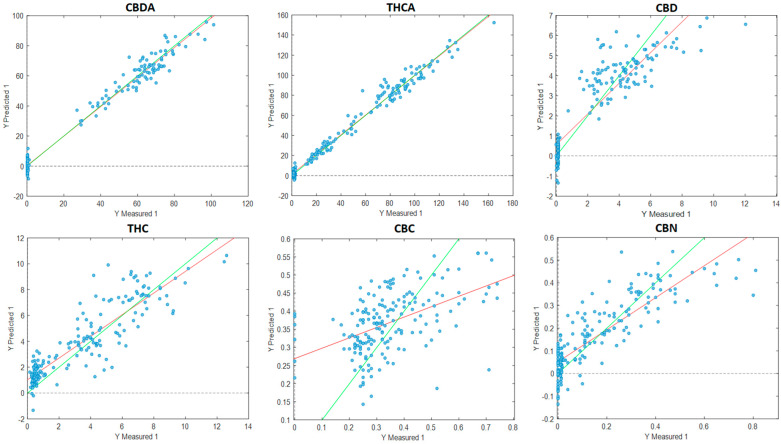
Plot of measured values versus the predicted values of cannabinoids in the validation dataset with data obtained from the VIAVI MicroNIR instrument processed from the PLS_toolbox using the PLS-R tool. Cannabidiolic acid (CBDA), tetrahydrocannabinolic acid (THCA-A), cannabidiol (CBD), tetrahydrocannabinol (THC), cannabichromene (CBC), cannabinol (CBN), cannabidivaric acid (CBDVA), cannabidivarin (CBDV), cannabigerolic acid (CBGA), cannabigerol (CBG), tetrahydrocannabidivarin (THCV), tetrahydrocannabidivarinic acid (THCVA), cannabinolic acid (CBNA) and cannabichromenic acid (CBCA). Units are expressed in mg/g. Green line: line of best fit from reference data; red line: line of best fit from predicted data.

**Table 1 sensors-23-02607-t001:** Calibration and validation values of partial least square discriminant analysis (PLS-DA) models on the ratio of cannabinoid compounds analysed by the Bruker MPA II instrument.

	Even Ratio	High CBDA	High THCA
Sensitivity (Cal)	0.942	1	1
Specificity (Cal)	0.981	0.995	1
Sensitivity (CV)	0.942	1	1
Specificity (CV)	0.981	0.995	1
Sensitivity (Pred)	1	1	1
Specificity (Pred)	0.987	1	1
Class. Err (Cal)	0.038	0.002	0
Class. Err (CV)	0.038	0.002	0
Class. Err (Pred)	0.007	0	0
RMSEC	0.237	0.148	0.123
RMSECV	0.248	0.157	0.126
RMSEP	0.211	0.127	0.106
Bias	1.11 × 10^−16^	−2.50 × 10^−16^	−1.11 × 10^−16^
CV Bias	−0.001	0.001	0.000
Pred Bias	0.066	−0.040	−0.027
*R* ^2^ _Cal_	0.739	0.881	0.939
*R* ^2^ _CV_	0.714	0.867	0.936
*R* ^2^ _Pred_	0.737	0.936	0.957

CBDA: cannabidiolic acid; THCA: tetrahydrocannabinolic acid; Class. Err.: classification error; RMSEC: root-mean-square error of calibration; RMSECV: root-mean-square error of cross-validation; RMSEP: root-mean-square error of prediction; CV Bias: cross-validation bias; Pred Bias: calculated prediction bias; *R*^2^_Cal_: coefficient of determination of calibration; *R*^2^_CV_: coefficient of determination of cross-validation; *R*^2^_Pred_: coefficient of regression of measured data vs. predicted data. High CBDA > 6:1; high THCA < 1:6; 6:1 > even ratio > 1:6 (x:x = CBDA:THCA ratio). Permutation testing (*n* = 50); returned *p*-value < 0.05.

**Table 2 sensors-23-02607-t002:** Calibration and validation values of partial least square discriminant analysis (PLS-DA) models on the ratio of cannabinoid compounds analysed by the Bruker MPA II instrument.

	Even Ratio	High CBDA	High THCA
Sensitivity (Cal)	0.779	0.987	0.991
Specificity (Cal)	0.725	0.887	1
Sensitivity (CV)	0.779	0.987	0.987
Specificity (CV)	0.717	0.889	1
Sensitivity (Pred)	0.907	0.979	1
Specificity (Pred)	0.755	0.91	1
Class. Err (Cal)	0.248	0.063	0.004
Class. Err (CV)	0.252	0.062	0.006
Class. Err (Pred)	0.169	0.055	0
RMSEC	0.391	0.305	0.165
RMSECV	0.394	0.306	0.167
RMSEP	0.368	0.298	0.125
Bias	0	5.55 × 10^−17^	1.11 × 10^−16^
CV Bias	0.000	0.000	2.82 × 10^−5^
Pred Bias	0.023	−0.007	−0.016
*R* ^2^ _Cal_	0.267	0.535	0.889
*R* ^2^ _CV_	0.256	0.531	0.885
*R* ^2^ _Pred_	0.253	0.544	0.939

CBDA: cannabidiolic acid; THCA: tetrahydrocannabinolic acid; Class. Err.: classification error; RMSEC: root-mean-square error of calibration; RMSECV: root-mean-square error of cross-validation; RMSEP: root-mean-square error of prediction; CV Bias: cross-validation bias; Pred Bias: calculated prediction bias; *R*^2^_Cal_: coefficient of determination of calibration; *R*^2^_CV_: coefficient of determination of cross-validation; *R*^2^_Pred_: coefficient of regression of measured data vs. predicted data. High CBDA > 6:1; high THCA < 1:6; 6:1 > even ratio > 1:6 (x:x = CBDA:THCA ratio). Permutation testing (*n* = 50); returned *p*-value < 0.05.

**Table 3 sensors-23-02607-t003:** The different pre-processing methods applied to partial least squares regression models (PLS-R) for the determination of cannabinoid compounds using data from the Bruker MPA instrument.

	Region (cm^−1^)	Scatter Correction	Derivative	LV	N	RMSEC (mg/g)	*R* ^2^ _Cal_	RMSECV (mg/g)	*R* ^2^ _CV_	RMSEP (mg/g)	Pred Bias	*R* ^2^ _Pred_
CBDA	9000–4000	DT, SNV and MC	2, 2, 5	2	734	6.93	0.96	7.00	0.96	4.79	0.38	0.98
THCA	9000–4000	DT, SNV and MC	2, 2, 5	6	734	5.38	0.99	5.59	0.98	5.51	−0.28	0.98
CBD	9000–4000	DT, SNV and MC	2, 2, 5	8	734	0.96	0.85	1.10	0.80	1.02	−0.09	0.89
THC	9000–4000	DT, SNV and MC	2, 2, 5	8	734	1.29	0.86	1.42	0.83	0.93	0.22	0.93
CBC	9000–4000	DT, SNV and MC	2, 2, 5	8	734	0.14	0.45	0.34	0.34	0.11	0.00001	0.46
CBN	9000–4000	DT, SNV and MC	2, 2, 15	8	734	0.10	0.75	0.10	0.72	0.08	−0.001	0.80
CBDVA	9000–4000	DT, SNV and MC	2, 2, 5	7	734	0.88	0.63	0.93	0.58	0.08	−0.04	0.60
CBDV (*)	9000–4000	DT, SNV and MC	2, 2, 5	5	734	0.08	0.55	0.08	0.49	0.59	0.00	0.59
CBGA	9000–4000	DT, SNV and MC	2, 2, 15	9	734	1.03	0.54	1.12	0.47	0.87	−0.01	0.55
CBG	9000–4000	DT, SNV and MC	2, 2, 7	7	734	0.24	0.49	0.26	0.39	0.23	0.01	0.37
THCV	9000–4000	DT, Norm and MC	2, 2, 7	10	734	0.07	0.45	0.08	0.34	0.06	−0.001	0.44
THCVA	9000–4000	DT, SNV and MC	2, 2, 7	3	734	0.80	0.28	0.82	0.25	0.86	−0.07	0.34
CBNA	9000–4000	DT, SNV and MC	2, 2, 3	9	734	0.20	0.85	0.23	0.80	0.26	−0.00003	0.80
CBCA	9000–4000	DT, SNV and MC	2, 2, 15	4	734	0.67	0.59	0.69	0.57	0.63	0.15	0.61

CBDA: cannabidiolic acid; THCA-A: tetrahydrocannabinolic acid; CBD: cannabidiol; THC: tetrahydrocannabinol; CBC: cannabichromene; CBN: cannabinol; CBDVA: cannabidivaric acid; CBDV: cannabidivarin; CBGA: cannabigerolic acid; CBG: cannabigerol; THCV: tetrahydrocannabidivarin; THCVA: tetrahydrocannabidivarinic acid; CBNA: cannabinolic acid; CBCA: cannabichromenic acid; DT: detrend; SNV: standard normal variate; Norm: normalisation; MC: mean centring; derivative pre-treatment: the first digit is the polynomial order, the second digit is the derivative order and the third digit is the datapoint gap in which the derivative is calculated; LV: number of latent variables; N: number of unique samples; RMSEC: root-mean-square error of calibration; *R*^2^_Cal_: coefficient of determination of calibration; RMSECV: root-mean-square error of cross-validation; *R*^2^_CV_: coefficient of determination of cross-validation; RMSEP: root-mean-square error of prediction; Pred Bias: calculated prediction bias; *R*^2^_Pred_: coefficient of regression of measured data vs. predicted data. Permutation testing (*n* = 50); returned *p*-value < 0.05. (*) Sample population bias.

**Table 4 sensors-23-02607-t004:** The different pre-processing methods applied to partial least squares regression models (PLS-R) for the determination of cannabinoid compounds using data from the VIAVI MicroNIR Onsite-W instrument.

	Region (cm^−1^)	Scatter Correction	Derivative	LV	N	RMSEC (mg/g)	*R* ^2^ _Cal_	RMSECV (mg/g)	*R* ^2^ _CV_	RMSEP (mg/g)	Pred Bias	*R* ^2^ _Pred_
CBDA	10526–6060	DT, SNV and MC	2, 2, 5	2	730	6.41	0.97	6.53	0.96	4.75	−0.25	0.98
THCA	10526–6060	DT, SNV and MC	2, 2, 5	5	730	5.98	0.98	6.45	0.98	6.23	−0.13	0.98
CBD	10526–6060	DT, SNV and MC	2, 2, 5	5	730	1.28	0.75	1.36	0.72	1.15	−0.04	0.80
THC	10526–6060	DT, SNV and MC	2, 2, 5	7	730	1.71	0.75	1.83	0.72	1.66	−0.01	0.75
CBC	10526–6060	DT, SNV and MC	2, 2, 5	6	730	0.15	0.34	0.16	0.26	0.13	0.02	0.25
CBN	10526–6060	DT, SNV and MC	2, 2, 5	9	730	0.11	0.66	0.12	0.60	0.10	0.00	0.71
CBDVA	10526–6060	DT, SNV and MC	2, 2, 5	4	730	1.02	0.49	1.06	0.45	1.07	−0.13	0.55
CBDV (*)	10526–6060	DT, SNV and MC	2, 2, 5	3	730	0.08	0.47	0.08	0.43	0.08	0.00	0.51
CBGA	10526–6060	DT, SNV and MC	2, 2, 5	9	730	1.08	0.53	1.17	0.45	0.92	0.08	0.38
CBG	10526–6060	DT, SNV and MC	2, 2, 5	6	730	0.23	0.47	0.24	0.40	0.28	0.00	0.34
THCV	10526–6060	DT, SNV and MC	2, 2, 5	3	730	0.08	0.32	0.08	0.23	0.09	0.00	0.21
THCVA	10526–6060	DT, SNV and MC	2, 2, 5	11	730	0.80	0.36	0.90	0.21	0.75	0.01	0.28
CBNA	10526–6060	DT, SNV and MC	2, 2, 5	8	730	0.29	0.71	0.31	0.67	0.24	0.01	0.76
CBCA	10526–6060	DT, SNV and MC	2, 2, 5	6	730	0.69	0.59	0.72	0.55	0.56	0.07	0.66

CBDA: cannabidiolic acid; THCA-A: tetrahydrocannabinolic acid; CBD: cannabidiol; THC: tetrahydrocannabinol; CBC: cannabichromene; CBN: cannabinol; CBDVA: cannabidivaric acid; CBDV: cannabidivarin; CBGA: cannabigerolic acid; CBG: cannabigerol; THCV: tetrahydrocannabidivarin; THCVA: tetrahydrocannabidivarinic acid; CBNA: cannabinolic acid; CBCA: cannabichromenic acid; DT: detrend; SNV: standard normal variate; Norm: normalisation; MC: mean centring; derivative pre-treatment: the first digit is the polynomial order, the second digit is the derivative order and the third digit is the datapoint gap in which the derivative is calculated; LV: number of latent variables; N: number of unique samples; RMSEC: root-mean-square error of calibration; *R*^2^_Cal_: coefficient of determination of calibration; RMSECV: root-mean-square error of cross-validation; *R*^2^_CV_: coefficient of determination of cross-validation; RMSEP: root-mean-square error of prediction; Pred Bias: calculated prediction bias; *R*^2^_Pred_: coefficient of regression of measured data vs. predicted data. Permutation testing (*n* = 50); returned *p*-value < 0.05. (*) Sample population bias.

**Table 5 sensors-23-02607-t005:** Correlation matrix of 14 cannabinoids taken from all harvest groups.

	CBDA	CBD	CBN	THC	CBC	THCA	CBDVA	CBDV	CBGA	CBG	THCV	THCVA	CBNA	CBCA
CBDA	1.00	0.79	−0.55	−0.63	0.21	−0.80	0.57	0.60	−0.22	−0.30	−0.36	−0.44	−0.71	0.66
CBD	0.79	1.00	−0.40	−0.50	0.44	−0.76	0.53	0.72	−0.26	−0.27	−0.27	−0.39	−0.64	0.41
CBN	−0.55	−0.40	1.00	0.78	0.17	0.45	−0.41	−0.38	0.06	0.18	0.45	0.26	0.69	−0.24
THC	−0.63	−0.50	0.78	1.00	0.13	0.66	−0.52	−0.49	0.27	0.40	0.53	0.43	0.56	−0.23
CBC	0.21	0.44	0.17	0.13	1.00	−0.22	0.26	0.36	−0.04	0.00	0.11	−0.09	−0.09	0.44
THCA	−0.80	−0.76	0.45	0.66	−0.22	1.00	−0.54	−0.60	0.43	0.52	0.34	0.50	0.70	−0.31
CBDVA	0.57	0.53	−0.41	−0.52	0.26	−0.54	1.00	0.82	−0.01	−0.13	−0.14	−0.20	−0.50	0.37
CBDV	0.60	0.72	−0.38	−0.49	0.36	−0.60	0.82	1.00	−0.18	−0.19	−0.13	−0.26	−0.50	0.35
CBGA	−0.22	−0.26	0.06	0.27	−0.04	0.43	−0.01	−0.18	1.00	0.46	0.09	0.15	0.18	0.03
CBG	−0.30	−0.27	0.18	0.40	0.00	0.52	−0.13	−0.19	0.46	1.00	0.17	0.20	0.24	−0.06
THCV	−0.36	−0.27	0.45	0.53	0.11	0.34	−0.14	−0.13	0.09	0.17	1.00	0.80	0.32	−0.17
THCVA	−0.44	−0.39	0.26	0.43	−0.09	0.50	−0.20	−0.26	0.15	0.20	0.80	1.00	0.34	−0.22
CBNA	−0.71	−0.64	0.69	0.56	−0.09	0.70	−0.50	−0.50	0.18	0.24	0.32	0.34	1.00	−0.29
CBCA	0.66	0.41	−0.24	−0.23	0.44	−0.31	0.37	0.35	0.03	−0.06	−0.17	−0.22	−0.29	1.00

Values nearing 1.00 indicate a positive correlation; −1.00 indicates a negative correlation; and 0 indicates no correlation. Cannabidiolic acid (CBDA), tetrahydrocannabinolic acid (THCA-A), cannabidiol (CBD), tetrahydrocannabinol (THC), cannabichromene (CBC), cannabinol (CBN), cannabidivaric acid (CBDVA), cannabidivarin (CBDV), cannabigerolic acid (CBGA), cannabigerol (CBG), tetrahydrocannabidivarin (THCV), tetrahydrocannabidivarinic acid (THCVA), cannabinolic acid (CBNA) and cannabichromenic acid (CBCA).

**Table 6 sensors-23-02607-t006:** Calibration and validation values of partial least squares regression models on finely ground cannabinoid compounds analysed by the Bruker MPA II instrument.

	Region (cm^−1^)	Scatter Correction	Derivative	LV	N	RMSEC (mg/g)	*R* ^2^ _Cal_	RMSECV (mg/g)	*R* ^2^ _CV_	RMSEP (mg/g)	Pred Bias	*R* ^2^ _Pred_
CBDA	9000–4000	DT, SNV and MC	2, 2, 15	4	479	3.93	0.98	4.20	0.98	2.64	−0.23	0.99
THCA	9000–4000	DT, SNV and MC	2, 2, 3	8	479	4.31	0.99	4.99	0.98	5.26	−0.30	0.98
CBD	9000–4000	DT, SNV and MC	2, 2, 5	5	479	0.89	0.80	0.96	0.77	0.74	0.02	0.88
THC	9000–4000	DT, SNV and MC	2, 2, 5	9	479	0.82	0.92	0.99	0.89	0.80	0.07	0.91
CBC	9000–4000	DT, SNV and MC	2, 2, 5	4	479	0.15	0.25	0.16	0.18	0.13	0.02	0.28
CBN	9000–4000	DT, SNV and MC	2, 2, 15	10	479	0.08	0.80	0.09	0.75	0.11	−0.01	0.71
CBDVA	9000–4000	DT, SNV and MC	2, 2, 5	2	479	0.25	0.39	0.26	0.37	0.14	0.01	0.44
CBDV (*)	9000–4000	DT, SNV and MC	2, 2, 3	2	479	0.02	0.85	0.02	0.82	0.01	0.00	0.93
CBGA	9000–4000	DT, SNV and MC	2, 2, 5	5	479	1.15	0.48	1.22	0.42	1.08	0.07	0.43
CBG	9000–4000	DT, SNV and MC	2, 2, 15	4	479	0.26	0.40	0.27	0.36	0.26	−0.01	0.28
THCV	9000–4000	DT, SNV and MC	2, 2, 15	3	479	0.10	0.13	0.10	0.09	0.09	0.00	0.14
THCVA	9000–4000	DT, SNV and MC	2, 2, 15	2	479	0.85	0.24	0.86	0.23	0.84	0.18	0.09
CBNA	9000–4000	DT, SNV and MC	2, 2, 5	7	479	0.19	0.85	0.21	0.81	0.19	−0.02	0.79
CBCA	9000–4000	DT, SNV and MC	2, 2, 5	4	479	0.70	0.53	0.73	0.48	0.60	0.03	0.60

CBDA: cannabidiolic acid; THCA-A: tetrahydrocannabinolic acid; CBD: cannabidiol; THC: tetrahydrocannabinol; CBC: cannabichromene; CBN: cannabinol; CBDVA: cannabidivaric acid; CBDV: cannabidivarin; CBGA: cannabigerolic acid; CBG: cannabigerol; THCV: tetrahydrocannabidivarin; THCVA: tetrahydrocannabidivarinic acid; CBNA: cannabinolic acid; CBCA: cannabichromenic acid; DT: detrend; SNV: standard normal variate; Norm: normalisation; MC: mean centring; derivative pre-treatment: the first digit is the polynomial order, the second digit is the derivative order and the third digit is the datapoint gap in which the derivative is calculated; LV: number of latent variables; N: number of unique samples; RMSEC: root-mean-square error of calibration; *R*^2^_Cal_: coefficient of determination of calibration; RMSECV: root-mean-square error of cross-validation; *R*^2^_CV_: coefficient of determination of cross-validation; RMSEP: root-mean-square error of prediction; Pred Bias: calculated prediction bias; *R*^2^_Pred_: coefficient of regression of measured data vs. predicted data. Permutation testing (*n* = 50); returned *p*-value < 0.05. (*) Sample population bias.

**Table 7 sensors-23-02607-t007:** Calibration and validation values of partial least squares regression models on coarsely ground cannabinoid compounds analysed by the Bruker MPA II instrument.

	Region (cm^−1^)	Scatter Correction	Derivative	LV	N	RMSEC (mg/g)	*R* ^2^ _Cal_	RMSECV (mg/g)	*R* ^2^ _CV_	RMSEP (mg/g)	Pred Bias	*R* ^2^ _Pred_
CBDA	9000–4000	DT, SNV and MC	2, 2, 3	4	479	1.95	0.97	2.02	0.97	1.64	−0.27	0.98
THCA	9000–4000	DT, SNV and MC	2, 2, 5	5	479	9.64	0.94	10.31	0.93	9.42	−1.86	0.94
CBD	9000–4000	DT, SNV and MC	2, 2, 3	6	479	1.02	0.74	1.17	0.66	0.87	0.07	0.80
THC	9000–4000	DT, SNV and MC	2, 2, 3	6	479	1.11	0.86	1.23	0.83	1.06	0.18	0.83
CBC	9000–4000	DT, SNV and MC	2, 2, 5	6	479	0.14	0.37	0.16	0.28	0.11	0.02	0.34
CBN	9000–4000	DT, SNV and MC	2, 2, 5	9	479	0.08	0.83	0.09	0.77	0.10	0.00	0.69
CBDVA	9000–4000	DT, SNV and MC	2, 2, 3	5	479	0.22	0.55	0.25	0.45	0.10	0.00	0.53
CBDV (*)	9000–4000	DT, SNV and MC	2, 2, 3	4	479	0.02	0.83	0.02	0.81	0.01	0.00	0.95
CBGA	9000–4000	DT, SNV and MC	2, 2, 5	10	479	0.79	0.75	1.01	0.60	1.01	−0.13	0.51
CBG	9000–4000	DT, SNV and MC	2, 2, 5	7	479	0.24	0.45	0.25	0.40	0.24	−0.01	0.51
THCV	9000–4000	DT, SNV and MC	2, 2, 5	8	479	0.07	0.47	0.08	0.25	0.09	0.00	0.43
THCVA	9000–4000	DT, SNV and MC	2, 2, 15	8	479	0.72	0.42	0.80	0.30	0.74	0.13	0.37
CBNA	9000–4000	DT, SNV and MC	2, 2, 3	9	479	0.18	0.86	0.21	0.79	0.19	0.03	0.79
CBCA	9000–4000	DT, SNV and MC	2, 2, 11	5	479	0.70	0.52	0.73	0.48	0.63	0.07	0.59

CBDA: cannabidiolic acid; THCA-A: tetrahydrocannabinolic acid; CBD: cannabidiol; THC: tetrahydrocannabinol; CBC: cannabichromene; CBN: cannabinol; CBDVA: cannabidivaric acid; CBDV: cannabidivarin; CBGA: cannabigerolic acid; CBG: cannabigerol; THCV: tetrahydrocannabidivarin; THCVA: tetrahydrocannabidivarinic acid; CBNA: cannabinolic acid; CBCA: cannabichromenic acid; DT: detrend; SNV: standard normal variate; Norm: normalisation; MC: mean centring; derivative pre-treatment: the first digit is the polynomial order, the second digit is the derivative order and the third digit is the datapoint gap in which the derivative is calculated; LV: number of latent variables; N: number of unique samples; RMSEC: root-mean-square error of calibration; *R*^2^_Cal_: coefficient of determination of calibration; RMSECV: root-mean-square error of cross-validation; *R*^2^_CV_: coefficient of determination of cross-validation; RMSEP: root-mean-square error of prediction; Pred Bias: calculated prediction bias; *R*^2^_Pred_: coefficient of regression of measured data vs. predicted data. Permutation testing (*n* = 50); returned *p*-value < 0.05. (*) Sample population bias.

## Data Availability

Raw data are available upon request.

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
