# Peer review of "Developing Prediction Models Using Near-Infrared Spectroscopy to Quantify Cannabinoid Content in Cannabis Sativa"

_sensors, 2023, doi:10.3390/s23052607_

Round 1

Reviewer 1 Report

The manuscript shows a good comparison between different NIR spectrophotometers application, improving a rapid and affordable method to assess cannabinoid content with different exploring, classification and regression models. The large dataset analysed prevents population bias and permits to study and quantify the chemical variations. The comparison citations for each application also permits to better understand the NIRs advantages and the good results obtained in this research. Some little observations could be detected in the first two parts of manuscript (introduction and materials and methods), while in the last part there are not important observations (comments in the attached file).

Reviewer 2 Report

Tran et al. present a large and well-organized work on the use of NIR spectroscopy for both discrimination and determination of cannabinoid content in Cannabis sativa. Overall, it is an excellent work with a lot of workload and supporting dada in the supplementary files. The number samples used is huge and the discussion in some sections, such as from line 374 and in other sections, very good.

However, for this reviewer, the most important thing in this paper on the application of NIR spectroscopy is the quantification of the content of the different cannabinoids. Here I have some criticisms on the work. Authors based their discussion on the goodness of the models on reference 28 (a scientific paper from 2019). Williams et al. (2019) are not the authors who proposed how classify models based on the R2, but Shenk and Westerhaus (Shenk, J.S.; Westerhaus, M. Calibration the ISI way. In Near Infrared Spectroscopy: The Future Waves; Davies, A.M.C., Williams, P.C., Eds.; NIR Publications: Chichester, UK, 1996; pp. 198–202. ISBN 978-0952866602). Based on these authors, R2 of calibration models must be higher than 0.9 to be used for prediction/determination of concentrations, parameters, etc. Most importantly, the standard error of prediction (SEP) must be the closer the standard error of laboratory (SEL). A good discussion on these parameters, including why others such as RPD should not be used, can be found in García Martín, J.F. Potential of Near-Infrared Spectroscopy for the Determination of Olive Oil Quality. Sensors 2022, 22, 2831 (https://doi.org/10.3390/s22082831), which could be used to enhance the discussion section of this paper (for more details about RPD, this link is also useful https://www.academia.edu/4303409/Why_you_dont_need_to_use_RPD).

Authors used the R2 value to conclude whether a model is good and can be used for routine analysis. I do not entirely agree with it. Only using R2 seems not to be enough. For example, just looking Fig. 4 it can be clearly seen that most of the compounds (CBC, THC, CBN, CBDVA, etc.) cannot be properly measured by NIR spectroscopy. In this Figure, is it good the PLS model and the predicted values for CBDA? Well, the sample concentrations range seems to be 0 – 90 mg/g and the SEP = 4.79 mg/g. For samples with CBDA concentrations of 70, 80 or 90 mg/g, a prediction of its concentration by NIR spectroscopy with a mean deviation of 4.79 units from the actual value can be acceptable, but for samples with concentrations close to 0, it is not acceptable. In this figure, it is clearly seen that the samples with CBDA concentrations close to 0 mg/g were bad predicted. Therefore, is it a good PLS calibration model? It is something that authors must decide and justify in the text. In the same figure, it is clear that a SEP value of 1.02 mg/g for CBD samples with concentrations between 0 and 8 mg/g is too big, so the prediction fails. Therefore, what this reviewer ask the authors is to provide the SEL of each compound to compare with SEP. If SEL were not calculated and cannot be now calculated, maybe authors could provide which are the equipment detection limits or minimum permitted errors required by hospitals or organizations that usually determine these cannabinoids, so readers can have an idea of the goodness of the proposed NIR-PLS calibration models and whether can be used in these facilities.

Other minor points:

What kind of samples were used for sections 3.3.1 and 3.3.2 (and therefore for Table 3 and Table 4)? For section 3.5 the type of samples is clear (finely ground and coarsely ground samples), but how samples were pre-treated in the other sections is unknown.

It would be extremely useful adding the number of samples (which is one of strongest points of the paper) and, mainly, the range of concentrations of the samples and units of each cannabinoid compound. Without these ranges, SEP and SECV values do mean nothing.

I agree with authors that why sometimes models built with coarsely material works better than with finely ground samples deserves be researched to evaluate which kind of samples are more suitable for NIR spectroscopy. Obviously, this is out of the scope of the present paper, so I hope authors will investigate this and present the results in future publications.

Reviewer 3 Report

In this manuscript, the authors used NIR spectroscopy to quantify the cannabinoids in cannabis by benchtop NIR spectrometera and handheld NIR spectrometer, and the effect of coarse and fine grinding on the detection results was analyzed. In principle I recommend the publication of the manuscript if the authors can address the following comments:

1. The authors give the raw data but lack statistical analysis of the calibration and prediction group data. It is suggested to add it.

2. Lack of units in figures and tables.

3. Please check the appropriateness of the PLS-model in the manuscript, R2Pred greater than R2Cal may represent not an optimal model.

Reviewer 4 Report

In this paper, a quantitative analysis model for predicting cannabinoid content was developed by combining high quality liquid chromatography-mass spectrometry (LCMS-RRB- data with near-infrared spectroscopy data, the accuracy of content prediction by different instruments and processing methods was compared. This is the actual production needs, there are some problems need to be modified:

1. The introduction is well written, but there is a little problem. The theme of this paper is to predict the content of cannabinoids by combining near infrared spectroscopy with high quality liquid chromatography mass spectroscopy (LCMS) data, which can appropriately increase the references of similar research methods in other species and the advantages of this instrument to increase the persuasiveness of the experimental results in this paper.line 70, mode” should be plural.

2. We can not understand the significance of the biological duplication of chemical variables in lines 276 and 278.

3. In lines 276 and 277, how explain the absence of genetic variation through biological duplication of chemical variables?

4. There is an error in punctuation in sentence " According to Williams et al.,28 (2019) these values are still adequate for " in line 360.

5. In line 567, the author points out that coarse ground hemp can also meet the quality assurance standard. However, in line 536, the R2 of "CBC, THCVA,THCV" is less than 0.5, so I hope to explain it to increase its persuasiveness.

In line 567, the authors note that coarse hemp can also meet QA standards, so is there any standard or specification for the granularity range of coarse hemp?

Round 2

Reviewer 4 Report

In this paper, a quantitative analysis model for predicting cannabinoid content was developed by combining high quality liquid chromatography-mass spectrometry (LCMS-RRB) data with near-infrared spectroscopy data, the accuracy of content prediction by different instruments and processing methods was compared. This is the actual production needs, there are some problems need to be modified:

1.      In the 29 lines of the abstract, the sentence "this study demostrates that for many applications a portable hand held device would provide accurate quantitative data" may be ambiguous, because the data provided by NIR spectroscopy can not be directly used for content prediction, but can only be quantitatively analyzed by combining other methods with chemical component content determination.

2.      Line 349 and line 372,inconsistent formatting of cited authors
